# From Normal to Abnormal: Transforming Medical Images with Diffusion Models for Dataset Balancing

**Martin Paulikat**[1,2]                                        ND297@UNI-HEIDELBERG.DE
**Lennart Nauschütte**[1,2]                                    LNAUSCHU@UNI-MUENSTER.DE
**Martin Simon Kalteis**[3]                                       S.KALTEIS@PAICON.COM
**Witali Aswolinskiy**[3]                                    W.ASWOLINSKIY@PAICON.COM
**Achim Schneider**[4]                                       MVZ.SCHNEIDER@ICLOUD.COM
**Hermann Bussmann**[1,2]                       HERMANN.BUSSMANN@MED.UNI-HEIDELBERG.DE
**Christian Aichmüller**[3]                                   C.AICHMUELLER@PAICON.COM
**Magnus von Knebel Doeberitz**[1,2]  MAGNUS.KNEBEL-DOEBERITZ@MED.UNI-HEIDELBERG.DE

[1] *Department of Applied Tumor Biology, Universitätsklinikum Heidelberg, Heidelberg, Germany*

[2] *Clinical Cooperation Unit F210, German Cancer Research Center (DKFZ), Heidelberg, Germany*

[3] *PAICON GmbH, Heidelberg, Germany*

[4] *MVZ im Fürstenbergkarree Berlin, Berlin, Germany*

## Abstract

Digital colposcopy relies on the accurate identification of high-grade lesions in cervical images. This study explores the use of diffusion models to address class imbalance, a common challenge in medical datasets. We propose a method that synthetically generates high-grade lesion features within normal cervical images. This method was evaluated on datasets from Berlin and Cambodia, the latter having a significant scarcity of high-grade lesions. Our approach successfully balanced the dataset and improved diagnostic performance by 5%.

**Keywords:** diffusion model, digital colposcopy, balancing, cervical screening

## 1. Introduction

Cervical cancer is the second most common cancer type among women in Cambodia and the second leading cause of cancer deaths in women between the ages of 15 and 44 years (Bruni et al., 2023) even while Cambodia has a low HPV prevalence of 5 % (Ueda et al., 2019). This discrepancy is largely due to the high number of untreated cases, as diagnostic tools and vaccines are expensive and often inaccessible in resource-limited settings. A promising solution is a neural network-based screening approach for digital colposcopy.

However, training a network on data from countries with a low HPV prevalence automatically leads to imbalanced datasets, where crucial high-grade lesion cases are scarce. These rare cases are vital for accurate detection and treatment as they have a high risk of progressing further to cervical cancer and imbalanced datasets can lead to biased models with poor performance on the underrepresented classes (Chawla et al., 2002).

There are a multitude of techniques to handle imbalanced data sets, like oversampling the underrepresented class, class weighted loss or undersampling (Chawla et al., 2002), but they are not without drawbacks. For example, oversampling the underrepresented class can lead to overfitting, compromising its ability to generalize to new data. Conversely,

undersampling the majority class risks losing valuable information, which is particularly critical in medical data sets that are often small by nature.

This paper proposes a novel approach utilizing diffusion models, initially introduced by (Ho et al., 2020), for class-relevant image modification. We leverage this technique to modify normal images to contain high grade lesion features and thereby creating a better balanced dataset that improves the models accuracy and generalizes well across various population groups.

## 2. Methods

**Data:** We collected our data in two different locations. The first data set, containing 636 colposcopy images, was collected in Berlin, Germany. We created two partitions: One for training the diffusion model and classifiers, containing 334 normal and low grade lesion images and 228 high grade lesion images. The other, a partition of 27 normal and low grade lesions and 37 high grade lesions was used for testing the classifiers. High grade lesion images from the Berlin training data set were annotated by a medical student and furthermore reviewed by a colposcopy expert with over 30 years of experience. Lesion severity data was extracted from the histopathological report. Our second data set of 1252 colposcopy images was collected in Cambodia and consists of 1218 normal and low grade lesion images and 34 high grade lesion images. 300 normal images of the Cambodia data set were transformed by our method into high grade lesion images leading to an additional modified data set containing 918 normal and low grade lesion images and 334 high grade lesion images.

**Diffusion:** Likewise to (Ho et al., 2020), we used the UNet architecture (Ronneberger et al., 2015) for the denoising process. In our method, the annotated parts were extracted, to which noise was continuously added, increasing over the epochs. These sections were then processed by the diffusion model, which aims to recreate the original images out of the noise, using the mean squared error between the original, unnoisy image and the model output image as loss function. The diffusion model was trained for 200 epochs, and as it was only trained to recreate lesions, it learned to transform noisy image sections to artificial high grade lesions. After training, we created the modified Cambodia data set by drawing masks manually to outline the regions of the images where lesions are likely to occur. Likewise to the training process, we extracted the masked part, added noise and transformed them to artificial high grade lesions.

**Classification:** We conducted ten training sessions for two ResNet-50 classifiers, each for 100 epochs, and computed the average output from these sessions. This approach ensures that the observed performance is not due to chance, confirming the reliability of the results. One of the classifiers was trained on the Berlin and the original datasets and the other one the Berlin and the modified Cambodia datasets, using ROC curves and AUC values against the Berlin test set.

## 3. Results

We trained a diffusion model in order to modify 300 normal images of the Cambodia data set to introduce artificially generated images of high grade lesions in the data. Figure 1 panel A shows three examples of the transformation from normal to high grade lesion. We calculated the ROC curves and the AUC of the classifier results, which are shown in figure 1 panel B. The AUC of the classifier trained on the unmodified Cambodia data set was 0.69, while the AUC of the classifier trained on the modified Cambodia data set was 0.74.

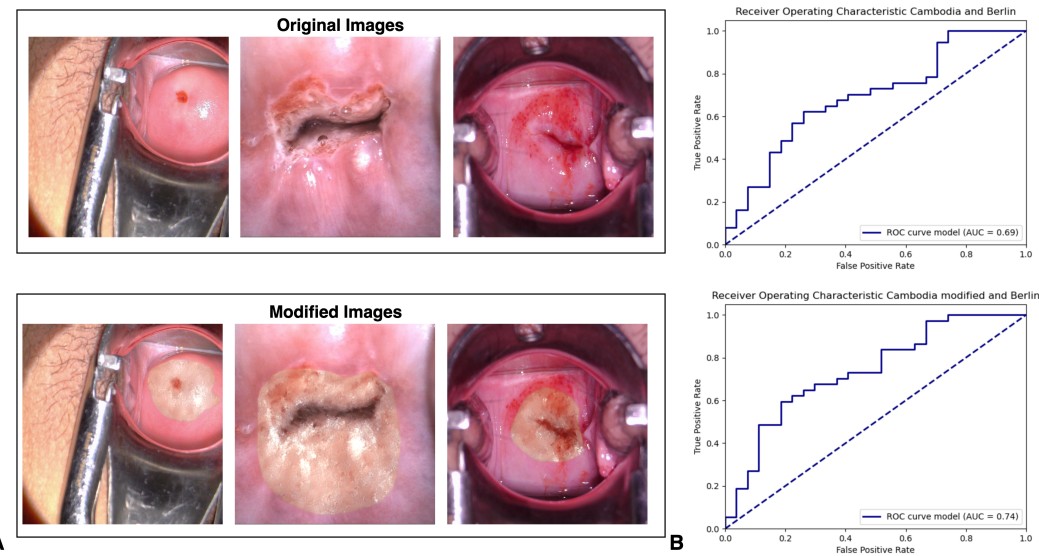

Figure 1: **A)** Example images taken from the Cambodia data set. The upper, normal images were modified by our method to contain high grade lesions, resulting into the lower images. **B)** ROC curves and AUC of the classifiers trained on the unmodified Cambodia data set (left) and the modified Cambodia data set (right) and tested on the Berlin test set. Both classifiers were also trained on the Berlin training set.

## 4. Discussion & Conclusion

The promising results of the first evaluation of our method indicate that it might alleviate the issue of imbalanced data sets in digital colposcopy. Furthermore, given the semi-supervised nature of our method, it can be assured that lesions are created in the correct location of the cervix. Future work will include applying the method to larger imbalanced datasets across various domains and modalities beyond digital colposcopy. Additionally, we will compare the performance of our method with other balancing techniques, such as weighted loss or weighted sampling, and determine whether it can effectively be combined with them to further enhance performance.

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
