# OpenReview forum: "From Normal to Abnormal: Transforming Medical Images with Diffusion Models for Dataset Balancing"
_MIDL.io/2024/Short_Papers — MIDL 2024 Short Papers_

### Official Review · Reviewer_FS86 · 2024-04-24

**Confidence:** 5
**Final Rating:** 5

**Review:**

This paper focuses on identification of high-grade lesions in cervical images. The main contribution of the paper is the investigation of a diffusion model to address dataset imbalance between normal/low-grade to high-grade lesions. The experimental design is rigorous with two datasets and ten replications to assess statistical significance. The results demonstrate potential of the approach.

---

### Decision · Program_Chairs · 2024-04-26

Accept